# Pharmacological Modulation of Temporal Discounting: A Systematic Review

**DOI:** 10.3390/healthcare11071046

**Published:** 2023-04-06

**Authors:** Luis Felipe Sarmiento, Jorge Alexander Ríos-Flórez, Hector Andres Paez-Ardila, Pêssi Socorro Lima de Sousa, Antonio Olivera-La Rosa, Anderson Manoel Herculano Oliveira da Silva, Amauri Gouveia

**Affiliations:** 1Laboratory of Neuroscience and Behavior, Federal University from Pará, Belém 66050-160, Brazil; 2Neuroanatomy Laboratory, Department of Morphology, Federal University of Rio Grande do Norte, Natal 59078-970, Brazil; 3Department of Psychology, Politécnico Grancolombiano University Institution, Medellín 745220, Colombia; 4Department of Psychology, Universidad Manuela Beltran, Bucaramanga 680004, Colombia; 5Department of Psychological and Social Sciences, Universidad Católica Luis Amigó, Medellín 050034, Colombia; 6Human Evolution and Cognition Group, University of the Balearic Islands, 07122 Palma de Mallorca, Spain; 7Laboratory of Experimental Neuropharmacology, Federal University from Pará, Belém 66050-160, Brazil

**Keywords:** temporal discounting, decision-making, intertemporal decisions, drug administration, impulsivity, self-control

## Abstract

Temporal discounting is a phenomenon where a reward loses its value as a function of time (e.g., a reward is more valuable immediately than when it delays in time). This is a type of intertemporal decision-making that has an association with impulsivity and self-control. Many pathologies exhibit higher discounting rates, meaning they discount more the values of rewards, such as addictive behaviors, bipolar disorder, attention-deficit/hyperactivity disorders, social anxiety disorders, and major depressive disorder, among others; thus, many studies look for the mechanism and neuromodulators of these decisions. This systematic review aims to investigate the association between pharmacological administration and changes in temporal discounting. A search was conducted in PubMed, Scopus, Web of Science, Science Direct and Cochrane. We used the PICO strategy: healthy humans (P-Participants) that received a pharmacological administration (I-Intervention) and the absence of a pharmacological administration or placebo (C-Comparison) to analyze the relationship between the pharmacological administration and the temporal discounting (O-outcome). Nineteen studies fulfilled the inclusion criteria. The most important findings were the involvement of dopamine modulation in a U-shape for choosing the delayed outcome (metoclopradime, haloperidol, and amisulpride). Furthermore, administration of tolcapone and high doses of d-amphetamine produced a preference for the delayed option. There was a time-dependent hydrocortisone effect in the preference for the immediate reward. Thus, it can be concluded that dopamine is a crucial modulator for temporal discounting, especially the D2 receptor, and cortisol also has an important time-dependent role in this type of decision. One of the limitations of this systematic review is the heterogeneity of the drugs used to assess the effect of temporal discounting.

## 1. Introduction

Intertemporal decision-making is choosing between different outcomes at different times [1]. A reward or outcome will lose value while time passes; this phenomenon is known as ‘temporal discounting’ or ‘delayed discounting’ [2]. As the value of the delayed in-time outcome is discounted, there is a higher bias for the sooner choice [3]. This decision is prevalent in daily life, specifically in the areas of health, education, investment, and even clinical conditions [4].

Temporal discounting is measured with tasks in which participants have to choose between a smaller amount of money delivered immediately or after a short amount of time and a large amount of cash given in the future. e.g., ‘Would you prefer 40 dollars today or 100 dollars in three months?’. A person has a higher discount rate when his or her preference goes towards the smaller and more immediate reward. On the other hand, the person has a lower discount rate when he has a stronger preference for the delayed and bigger reward [5].

Thus, the preference for the small, immediate reward over the higher, delayed one is considered an impulsive decision. The preference for the higher, delayed reward over the smaller, immediate reward is a self-controlled decision [6]. Non-human animals prefer impulsive decisions, such as monkeys, rats, mice, and pigeons [1]. The discounting is best described by a hyperbolic curve that shows how the reward loses value with the passage of time (Figure 1). Furthermore, the tasks that measure temporal discounting have a variable known as the k-value. It is a free parameter that measures the sensitivity to the delay. When the k value is low, it means the individual discount is lower with the delay in time, and when the k-value is high, the individual is susceptible to the delay [7].

This type of decision has higher discount rates in many pathologies such as addictive behavior [8], heroin addiction [9], cocaine addiction [10], alcohol addiction [11], opioid-dependents [12], smokers [13], bipolar disorder [14], attention-deficit/hyperactivity disorder [15], antisocial personality disorder, social anxiety disorder [16], borderline personality disorder [17], major depressive disorder [18], schizophrenia and schizoaffective disorders [19], pathological gambling, and orbitofrontal cortex lesions [20].

## 2. Methods

This systematic review followed the PRISMA guidelines [21]—[see Appendix A]. Moreover, it was registered in the PROSPERO database (https://www.crd.york.ac.uk/PROSPERO/ (accessed on 28 April 2020)), with the registration number CRD42020161785.

### 2.1. Eligibility Criteria and Search Strategy

The PICO strategy was used to identify relevant studies in the question being investigated. PICO is an acronym for Population (P), intervention (I), Comparison (C), and Outcome (O). This systematic review used Healthy Humans (P), Pharmacological administration (I), and Intertemporal decisions (O). The exclusion criteria considered review articles, study cases, book chapters, research in animals, letters to the editor, editorials, protocols, pilot studies, and studies without the information required in the question.

The guide question for this review was the following: Is temporal discounting affected by pharmacological interventions? The search was completed in various databases, including PubMed, Scopus, Web of Science, Cochrane, Science Direct, and Google Scholar. The Mesh terms, keywords, and search strategies were organized according to each database and are included in the Appendix A.

As of the start of the research, we had programmed an alert for new articles in each database. There were no time or language restrictions in the search for articles. Two reviewers performed the search and selection of the studies, the extraction of the data, and the risk of bias analyses as a double-blind evaluation. The references were uploaded into the citation manager (EndNote^®^, version X9, Thomson Reuters, Philadelphia, PA, USA, EUA).

### 2.2. Data Extraction

The EndNote^®^ removed the duplicate references first automatically, followed by the authors conducting it manually. The studies were screened by title and abstract. The remaining ones went through a full-text review, and the articles that did not fulfill the inclusion criteria were removed. From the selected studies, we extracted: authors, country, publication year, type of study design, participant characteristics (sample size), age, mean, intertemporal decision task, pharmacological kind of intervention/dose, and results (Table 1).

### 2.3. Data Analysis

We chose Rob 2.0—a revised Cochrane risk of bias tool for randomized trials—to evaluate the risk of bias. It is a reviewed tool used and adapted to assess different types of study interventions. The Rob 2.0 proposes a series of questions based on five domains of bias and an overall evaluation. The domains are deviation from intended intervention, missing outcome data, measurement of the outcome, selection of the reported result, and overall. Each domain presents some questions related to the trial design, conduction, and reporting of the results. The question is answered as Yes, Probably Yes, No, or Probably No. These responses stratify the domains as High Risk if there are significant problems, ‘Some concerns’ if there are minor problems, or ‘Low Risk’ if there are no problems.

## 3. Results

### 3.1. Selection of the Studies

The database search identified 2065 references, of which 477 were duplicate references. The remaining 1588 references were analyzed by title and abstract. Twenty-three articles fulfilled the inclusion criteria and went to full-text analysis.

From the twenty-three articles assessed by full-text analysis, eight were excluded, seven of them because they did not have a pharmacological intervention and one because it used rats as subjects. Finally, nineteen studies fulfilled the inclusion criteria and went through the quality review. Each step of the review was performed independently by two reviewers, and a third reviewer reviewed the included articles. The steps are described in the PRISMA flowchart (Figure 2).

### 3.2. Characteristics of the Studies

The study samples were healthy volunteers; participants were men and women aged between 18 and 45 years. The experimental designs of the studies were within-subject and placebo-controlled for eleven studies [13,22,25,28,29,31,32,33,35,36,42] and eight studies used between-subject placebo-controlled designs [2,26,27,30,34,38,39,41].

Temporal discounting was measured by different types of tasks, such as the Delay Discounting Task [13,22,25,28,29,33,42] developed by Richard et al. [23], The Monetary Choice Questionnaire [30,34,39] by Kirby et al. [9], The Probability Discounting Task developed by Madden et al. [4]; Wu et al. [41] used an Intertemporal Choice Task adapted from Shen et al. [43], and, other studies designed their types of temporal discounting tasks [2,26,27,31,32,35,38].

The studies used the following pharmacological administrations (Table 2): diazepam 5 mg and 10 mg [13] and 20 mg in the Acheson et al. study [22]. Pramipexole (0.25 mg and 0.50 mg) [29]; yohimbine (20 mg) [30]; bupropion (75 mg and 100 mg), D- amphetamine, (5 mg, 10 mg, and 20 mg) [25,28]. Tolcapone (200 mg) [31]; propranolol (80 mg) [32]. Hydrocortisone (10 mg) [2,27]. Metoclopramide (10 mg) [26]; testosterone (150 mg and 50 mg) [34,41]. Oxycodone (5 mg, 10 mg, and 20 mg) [42] and L-dopa (187.5 mg and 150 mg) [35,36]. Amisulpride (400 mg), naltrexone (50 mg) [39], and haloperidol (2 mg and 1.5 mg) [38].

The administration of tolcapone [31], high (20 mg) doses of d-amphetamine [29], metoclopramide [26], haloperidol [38], amisulpride [39], and metoclopramide [26] promoted preference for the delayed reward. Hydrocortisone [2,27], testosterone [41], and L-dopa [36] generated preference for the immediate reward. 

On the other hand, administration of diazepam, pramipexole, yohimbine, bupropion, d-amphetamine, marinols, oxycodone, testosterone, and naltrexone did not produce any effect in the intertemporal decisions [13,22,29,30,33,34,39,42]. 

### 3.3. Risk of Bias

The risk of bias tool was an adapted version of Rob 2.0 that evaluated five domains of the studies: randomization process, deviation from the intended intentions, missing outcome data, measurements of the outcome, and selection of the reported results. Most of the studies presented a low risk of bias in all the domains. However, one of the studies showed some concerns in one of the domains (Figure 3).

Thus, eighteen of the studies were evaluated as having low Risk of bias in all domains Acheson et al. [22], Acheson and de Wit [25], Arrondo et al. [26], Cornelisse et al. [27], de Wit et al. [28], Hamidovic et al. [29], Herman et al. [30], Kayser et al. [31], Lempert et al. [32], Riis-Vestergaard et al. [2], Ortner et al. [34], Petzold et al. [35], Pine et al. [36], Reynolds et al. [13], Wu et al. [41], Weber et al. [39], Wagner et al. [38], and Zacny and de Wit [42]. Nevertheless, McDonald et al. [33] reported some concerns of bias in the domain of missing outcome data because the data for all the participants regarding the temporal discounting task were not available for analysis.

## 4. Discussion

This systematic review aimed to understand the association between the administration of some drugs and temporal discounting (see Table 3). Thus, nineteen studies fulfilled the inclusion criteria of the systematic review, of which eighteen presented a low risk of bias. Five studies that administered d-amphetamine (20 mg) [28], tolcapone (200 mg) [31], metoclopramide (10 mg) [26], haloperidol (2 mg) [38], and amisulpride (400 mg) [39] reported an increment at choosing the delayed reward. Four studies with drugs administered, such as hydrocortisone 10 mg [2,32], androgel 150 mg [41], and l-dopa 150 mg [36], found an increase in the preference for the more immediate reward. The administration of haloperidol 1.5 mg [36], pramipexole 0.25 mg and 0.50 mg [29], L-dopa 187.5 mg [35], d-amphetamine 5 mg [25], 10 mg [28], bupropion 75 mg and 100 mg [35], Yohimbine 20 mg [30], Propranolol 80 mg [32], testogel 50 mg [34], Oxycodone 5 mg, 10 mg and 20 mg [42], naltrexone [39], Marniols 7.5 mg and 15 mg [33], and Diazepam 5 mg and 10 mg [13], 20 mg [22] had no effect.

A systematic review is a set of scientific strategies applied to synthesize and analyze the best evidence on a specific topic. It summarizes and compiles the scientific evidence and decreases the risk of bias in the studies [44]. To achieve this, a quality assessment using a specific tool according to the types of studies included in the review was performed [45]. The systematic review gives a clear view of a topic, helps make decisions related to health and research, establishes new policies, and plans future research [44].

### 4.1. Dopaminergic System

Dopamine is a crucial neurotransmitter for animals that have been involved in a variety of functions and behaviors. This neuromodulator has been strongly associated with impulsivity and clinical disorders involving impulsive behaviors. Preclinical and human studies suggest a crucial role for dopaminergic function in temporal discounting. Parkinson’s disease patients, a disease known for dopamine deficiency, have shown altered temporal discounting compared with healthy volunteers [46]. Some of the drugs that act in the dopaminergic system, such as Tolcapone and L-dopa from the included studies are employed in the treatment of Parkinson’s disease.

There is a consistent finding within the studies. The dopamine receptor D2 modulation is involved in choosing the later reward over the sooner reward [26,38,39]. Three drugs act on the D2 receptor: amisulpride [39], haloperidol [38], and metoclopramide [26]. This D2 receptor has been associated with impulsive behavior.

In mice, the absence of the D2 receptor increased impulsive behavior, while restoration of the expression of the D2 receptor decreased impulsivity [47]. Another study knocked down these receptors in rats and found that these knockdown rats had a higher preference for the smaller and immediate reward than the control rats in a delay-discounting task [48]. Furthermore, lower ventral striatal D2 density is associated with impulsivity and greater temporal discounting, i.e., preference for the sooner option [49].

Following this reasoning, a study with methamphetamine-dependent subjects found they had lower striatal dopamine D2/D3 receptor availability than healthy controls, and higher impulsiveness was related to it [50]. Furthermore, another study found that methamphetamine-dependent subjects with lower striatal D2/D3 receptors have steeper temporal discounting [51].

A meta-analysis confirmed that Parkinson‘s disease patients have a steeper discount than healthy controls [52], emphasizing the crucial role of dopamine in this type of decision. Additionally, a study found that Parkinson’s disease patients treated with dopaminergic drugs exhibited greater farsightedness in their choices than Parkinson’s disease patients not on medication [53].

Wagner et al. [38] and Pine et al. [36] used the same drug, Haloperidol, where Pine et al. [36] found no effect of the drug. It should be noted that this difference could be due to the differences between the doses used in both studies: 2 mg [38] and 1.5 mg [36]. It is possible that lower doses of this dopamine antagonist produce a different result and that the effect of the drug would be dependent on dopaminergic baseline levels. It is hypothesized that dopamine has an inverted U-shape function between dopamine and impulsivity, where both extremes are associated with the worst decision-making performance [54]. It is consistent with results from the Castrellon et al. [55] meta-analysis, which found the same results in rats.

Hamidovic et al. [29] administered Pramipexole, a dopamine D3 receptor agonist, and found the drug had no effect. This result suggests D3 is not related to temporal discounting, but, as the author noted, it remains to be seen if multiple doses over a long period of time or higher doses present a different effect. Interestingly, Parkinson´s disease patients treated with Pramipexole have shown a proportion of impulse-control disorder related to the D3 receptors [56]. This has been seen even in rats where the D3 receptor is related to preferring the larger and delayed reward [57].

### 4.2. D-Amphetamine

D-Amphetamine is a drug known to have a high potential for abuse but is also used in the treatment of adults with Attention-Deficit Hyperactivity Disorder ADHD [58]. It can produce a feeling of well-being and euphoria and has helpful behavioral effects on ADHD patients. It increases dopamine in the synapses by binding to the dopamine transporter DAT, reversing DAT function in the medial prefrontal cortex, and inhibiting dopamine uptake [59]. Bupropion is commonly used to decrease impulsivity [60]. Bupropion is a drug that improves impulse control and attention in some patient populations, including those with ADHD [61]. Bupropion is a norepinephrine/dopamine-reuptake inhibitor. The administration of Bupropion had no effect on temporal discounting [25], the same as the administration of d-Amphetamine [25] and a low dose (10 mg) of d-Amphetamine [28]. However, administration of a higher dose (20 mg) in the same sample produced a preference for the later reward [28]. The higher dose by de Wit et al. [28] and the dose used by Acheson et al. [25] were the same (20 mg). However, the results differ, making the relationship between the acute pharmacological relationship and the task unclear. This is highlighted even more with the decrease in impulsivity (e.g., Go/No-Go task, Stop Task) found as other impulsivity measures were applied [28]. This could be explained by the dopamine U-shape action, as has been observed in mice and rats after d-amphetamine administration, where the shifted their response toward the sooner choice [62,63].

Another possible hypothesis Maguire et al. [64] found was that the effects of amphetamine differ depending on the manner in which the delayed reward is presented. This suggests that the changes in the results of the studies and the diverse performance were in part due to the sensitivity to the reward delay. 

### 4.3. Levodopa (L-Dopa)

L-dopa is a dopamine precursor that passes the blood-brain barrier and converts it into dopamine. Dopamine is not able to pass the blood-brain barrier. This drug is often used in the treatment of Parkinson´s disease. The results of this review showed that Pine et al. [36] found that L-dopa had an effect on preference for a sooner reward, whereas Petzold et al. [35] did not find any effect.

A secondary result from Petzold et al. [35] analyzed that low-impulsive individuals exhibit a preference for the sooner reward, converging with Pine et al. [36] results, but more impulsive individuals showed the opposite effect. These could be explained by the dopamine baseline levels, because, as mentioned, before dopamine behaves in a U-shape. Another possible hypothesis to explain this different result could be the difference in sample size between the two studies; Petzold et al. [35] had a sample size of 87 participants, while Pine et al. [36] had 14 participants. Animal studies have found that L-dopa leads to impulsive-like behaviors [65] and increases impulsivity in Parkinson´s disease even when improving some cognitive tasks [66].

### 4.4. Tolcapone

Tolcapone is a drug that inhibits the enzyme catechol-O-methyl transferase (COMT). This drug is used in the treatment of Parkinson´s disease. By inhibiting COMT, the degradation of L-dopa is prevented, permitting higher concentrations to cross the blood-brain barrier, and become dopamine [67]. After the administration of Tolcapone, Kayser et al. [31] found an increased preference for the later reward.

Congruently with this result, the genotype for the enzyme COTM predicts impulsive choice behavior. Specifically, subjects carrying enzymatically fewer active alleles encoding the COMT gene show a decrease in choosing the immediate reward [68,69]. Tolcapone increases dopamine tone prefernetially in the frontal cortex, and COMT is mainly in charge of degrading dopamine in the frontal cortex. Thus, low dopamine levels in the frontal cortex may predispose to higher impulsivity. 

### 4.5. Hypothalamic-Pituitary-Adrenal Axis (HPA-Axis)

The HPA axis is a major neuroendocrine system that aims to maintain physiological homeostasis and modulates many important processes such as the stress response, metabolism, fertility, and immunity. The hypothalamus secretes the corticotropin-releasing hormone (CRH), which releases the Adrenocorticotropic hormone (ACTH) from the pituitary gland. This stimulates the production of corticosteroids from the adrenal gland. Cortisol is the main corticosteroid and a stress biomarker. Cornelisse et al. [27] and Riis-Vestergaard et al. [2] administered hydrocortisone and found that after 15 min there was a change in the preference for the sooner reward but not the later. These results suggest a link between HPA axis activation and the mentioned acute stress-induced studies [3,32], where similar results were found. Administration of exogenous cortisol and increasing levels of endogenous cortisol produce the same response in intertemporal decisions. It should be noted that this response is time-dependent, as measures taken 185 min later did not reveal the results. 

### 4.6. Sympathetic-Adreno-Medullar System (SAM-System)

Herman et al. [30] and Lempert et al. [32] administered drugs that modulate the SAM system. The former used yohimbine to stimulate the system, and the latter used propranol to suppress it. Both found no effect. The Sympathetic-Adreno-Medullar System is part of the sympathetic nervous system. It gives a quick physiological response to face the challenge of outside stimuli. This reaction is mediated by catecholamines, especially epinephrine and norepinephrine, and leads to an increase in the heart rate and blood pressure. It is also one of the major systems involved in the stress response, along with the Hyphotalamic-Pituitary-Adrenal axis. Some studies administering protocols that produce acute stress have found that stressed individuals prefer the earlier option over the delayed option [3,70]. These pharmacological results could suggest that the SAM-system is not the main reason for the choice selected, but the HPA-axis should be further investigated.

### 4.7. Testosterone

Testosterone is a major sex hormone in men and women [71]. The two studies that looked for the effects of testosterone in temporal discounting found different results. While Wu et al. [41] found a preference for the smaller reward, Ortner et al. [34] found no effect. These differences could be dose-dependent; Wu et al. [41] administered 150 mg of testosterone, whereas Ortner et al. [34] used 50 mg. A meta-analysis reported a positive association between circulating testosterone and impulsivity [72]. This relationship could be possible due to the effects of testosterone in the dopaminergic system, as testosterone receptors are in dopaminergic neurons that project to the ventral striatum [73].

### 4.8. Opioids and Endocannabinoids

Opioidergic and cannabinoid systems involved receptors located throughout the brain and body. The opioid system is involved in affective processing, pain, pleasure, and reward [74]. The endocannabinoid system is involved in regulating physiological and cognitive processes, appetite, pain sensation, mood, and the pharmacological effects of cannabis [75]. Two studies administered drugs that act in the opioid system [39,42], and one in the endocannabinoid system [33]. All these studies were looking for the effects of these systems that have been implicated in drug addiction. None of the studies found any effect of the drugs used in participants.

These findings in the opioid system differ from preclinical studies in rats where the opioid receptor agonist, morphine, increased impulsive behavior in a temporal discounting task [76,77]. This difference could be explained by considering the differences between the tasks measured in the clinical and preclinical models. In the rats’ studies, the rats performed the task when the animal was in a drugged state, whereas in humans, the measure could be after the drug’s effects have dissipated. Furthermore, it is possible the temporal discounting is unaffected by drug effects, even when they affect other types of impulsivity. Some other acute administrations of drugs have failed to affect temporal discounting, for example, alcohol [78] and benzodiazepines [13].

McDonald et al. [33] also found no effect after the administration of THC. It was thought that THC would have a relationship with impulsivity modulated by the activity of dopaminergic neurons. THC and other cannabinoids increase the activity of dopaminergic neurons [79]. Moreover, earlier research found that marijuana alters the perception of time [80]. The null findings in the McDonald et al. [33] study may be due to the participants being required to decide on a reward that will be obtained after the effect of the drug has dissipated, as with opioid drugs.

### 4.9. Diazepam

Diazepam is a benzodiazepine that increases the inhibitory effects of gamma-aminobutyric acid (GABA). Some of the GABA functions are sleep induction, memory, anxiety, and epilepsy. Diazepam, as well as other benzodiazepines, are used for the treatment of anxiety, as a muscle relaxant, as an anticonvulsant, and sometimes are abused [40]. Acheson et al. [22] and Reynolds et al. [13] found no effect of diazepam in temporal discounting. 

These results are divergent from the literature that suggests Diazepam affects impulsive behavior in human and non-human models [13,81]. Deakin et al. [82] found in healthy volunteers that after administration of 29 mg of diazepam there were disinhibitory cognitive effects. There are reports that benzodiazepines produce disinhibition and increase aggression [83], and even benzodiazepines such as flunitrazepam, increase impulsivity and aggression [84]. Interestingly, it appears that Diazepam just increases some specific types of impulsive behavior but not others. Acheson et al. [22] found that Diazepam affects the performance of impulsive behavior in the stop task and go/no-go task but not the temporal discounting task.

All the studies in this systematic review went through a risk of bias analysis. Eleven studies were within the design interventions, and eight studies were placebo-controlled interventions. That makes Rob 2.0 an effective tool for the evaluation of the risk of bias and methodological assessment. All the studies presented a low risk of bias, except McDonald et al. [33], which had some concerns about missing outcome data.

## 5. Limitations

The findings of this study have to be seen in light of some limitations. While this systematic review followed a pre-registry strategy, recommended guidelines, and used a theory-based interpretation of the results, there are some limitations and considerations in this study that could be addressed in future research. First, the drugs administered to assess the effect on temporal discounting were extensively different, both in type of drug and dosage. This led us to decide against conducting a quantitative assessment of the evidence (meta-analysis), which in turn poses a limitation to the confidence of our conclusions. 

Furthermore, the small sample size of the studies included and the use of a k-value for the analysis should be taken into consideration. A potential methodological issue may be present when using the k-value on temporal discounting as it may affect the results, as seen in McDonald et al. [33], where the sample strongly decreases due to not fitting the k values. 

## 6. Conclusions

This systematic review allowed us to observe the effect of different types of drugs on Temporal Discounting. It showed that some drugs could influence the intertemporal decision outcomes in humans, either by lowering or by augmenting them. There is also evidence for possible clinical pathways to prevent or treat impulsiveness. This review points out that drugs involved with the dopaminergic systems exhibited changes in behavior. Especially, receptors D2 and D3 are related to the choice of the later reward. Furthermore, drugs that involve hormonal systems such as testosterone and cortisol impact temporal discounting, leading to more impulsive choices. It’s also important to keep in mind that cortisol’s impact on temporal discounting is time-dependent. Further research will continue to help understand the underlying physiological and neurological mechanism of temporal discounting and how this can be affected by the intake of drugs that lead the subject to be more impulsive or self-controlled.

## Figures and Tables

**Figure 1 healthcare-11-01046-f001:**
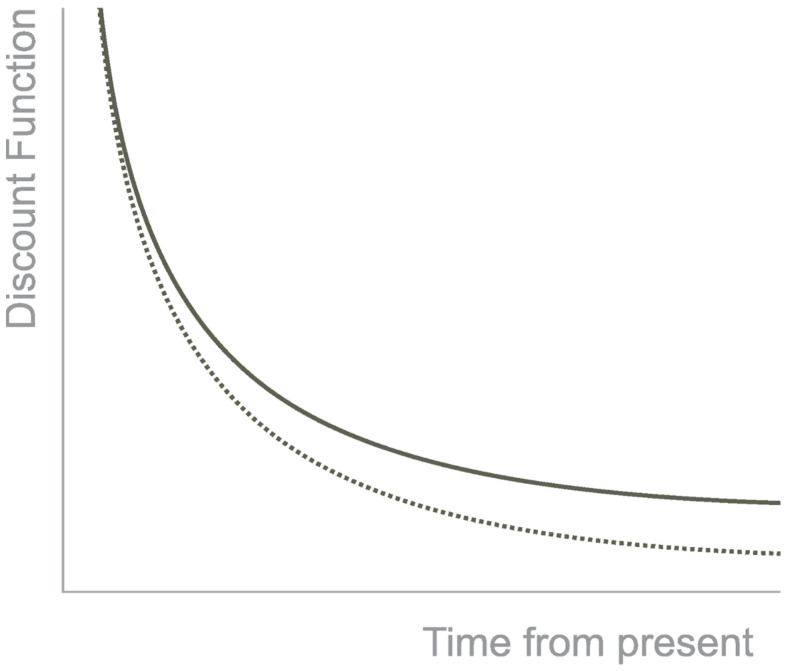
Hyperbolic Discounting. Note. The discounting of the outcome has a hyperbolic shape in function of time. The continuous line represents the hyperbolic function, and the dotted line an exponential function.

**Figure 2 healthcare-11-01046-f002:**
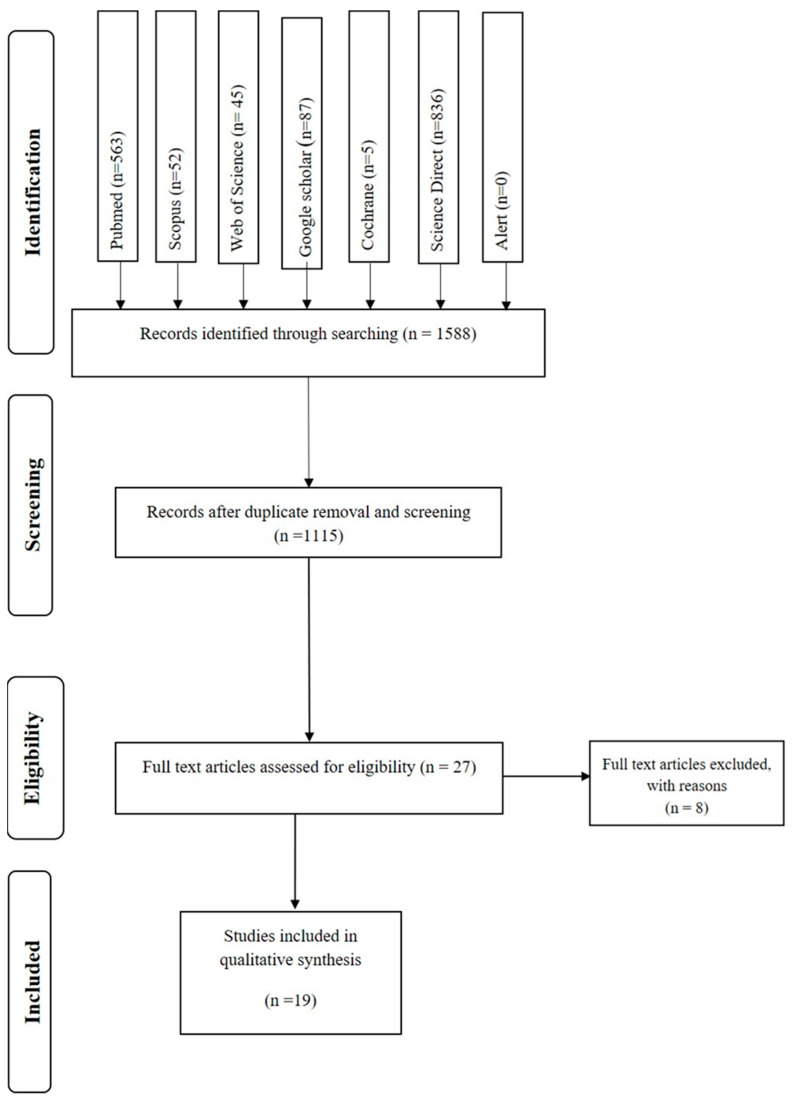
Prisma Flowchart. Note. Adapted PRISMA flow diagram of the study selection.

**Figure 3 healthcare-11-01046-f003:**
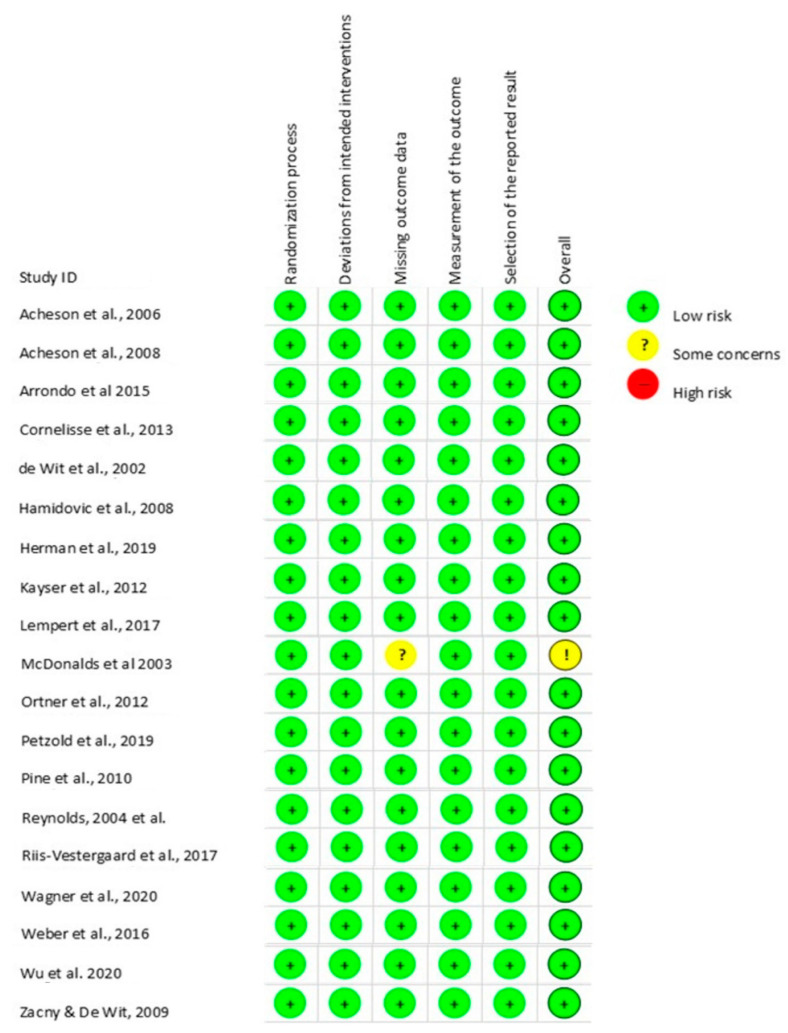
Analysis of Risk of bias. Note. Risk of bias using the Rob 2.0 tool for the 19 selected studies [2,13,22,25,26,27,28,29,30,31,32,33,34,35,36,38,39,41,42] (Risk +, ?, −), and the domains evaluated.

**Table 1 healthcare-11-01046-t001:** Summary of characteristics and results of included studies.

Author/Year/Study Design	Sample		Age (Mean ± SD), in Years	Intervention Type	Intertemporal Decision-Making Task	Results
	Source	Size				
Acheson et al. [22]. -United States -Randomized, within subject-design.	Males and females	18 participants -9 Male -9 Female	Aged: 18–45 Mean: 20.9 ± 2.5	Diazepam (Valium 20 mg). Placebo	Delay and probability discounting task [23]. Experiential discounting task [24].	Diazepam did not affect Delay and probability discounting task neither Experiential discounting task, *p* <0.05.
Acheson & de Wit [25]. -United States -Randomized, within-subject design.	Male and female	33 participants -Men:18 -Women:15	Aged: 18–45 Mean: 23.45 ± 4.6	Bupropion hydrochloride BH (75 mg) BH (100 mg) D-amphetamine sulfate (5 mg) Placebo	Delay and Probability Discounting Task [23].	Neither bupropion nor amphetamine affected discounting of hypothetical delayed or probabilistic rewards on the DPD, *p* <0.05.
Arrondo et al. [26]. -Spain -Randomized, between-subject design.	Males and Females	29 participants Intervention: 15 -Men:9 -Women:6 Placebo: 14 -Men:4 -Women:10	Aged intervention group: 24.0 ± 3.4 years Age placebo group: 23 ±1.9 years	Metoclopramide (10 mg) (Primperan) Placebo	Decision-making task.	Intervention group was more willing to wait to increase the probability of the reward, *p* = 0.007.
Cornelisse et al. [27]. - Germany - Randomized, between subject design.	Males	79 Participants	Aged 18 to 35	Hydrocortisone (10 mg). Placebo (albochin)	Temporal discounting task [27].	Hydrocortisone administration increased preference for immediate rewards after administration, but not three hours later. -*p* < 0.10 -*p* < 0.01 -*p* < 0.05
de Wit et al. [28] -United States -Randomized, within subject-subject design.	Males and females	36 participants -18 Male -18 Female	Aged 18–44 Mean: 24 ± 6.5	D-amphetamine (10 & 20 mg). Placebo	Delay Discounting [23]. Delay of Gratification. Time perception.	Only high d-Amphetamine significantly decreased the discounting, *p* < 0.05.
Hamidovic et al. [29]. -United States -Randomized, within design.	Males and females	10 Participants	Aged 18–28	Pramipexole (0.25 & 0.50 mg Mirapex) Placebo.	Delay Discounting task.	Pramipexole did not significantly affect either delay or probability discounting, *p* <0.05.
Herman et al. [30]. -UK -Randomized, between subject design.	Male and female	42 Participants Exposed: 21 -12 females -9 male Controls: 21 -11 females -10 Male	Aged 18–40 Exposed:21.29 Control: 23.19	Yohimbine (20 mg). Placebo	Probability Discounting Task (PD [4]). Monetary Choice Questionnaire (MCQ [4]).	There were no group differences in performance on neither MCQ, PD.
Kayser et al. [31]. -United States -Randomized, within-subject design.	Males and females	23 participants -13 females -10 males	Aged 19–41	Tolcapone (200 mg). Placebo	Delay Discounting Task.	Tolcapone (200 mg) increased the choosing of the delayed reward ∆ICR = −0.04 *p* = 0.025.
Lempert et al. [32]. -United States -Randomized, within-subject design.	Male and female	37 participants	Aged 27.8 ± 6.6	Propranolol (80 mg) Placebo	Intertemporal choice task.	Propranolol did not reduce temporal discounting overall, *p* < 0.05.
McDonald et al. [33]. -United States -Randomized, within-subjects design.	Males and females	37 participants -Men:18 -Women:19	Aged 18–45 Mean 23 ± 4.48	Marinols 7.5 mg Marinols 15 mg Placebo	The Delay discounting task [23].	THC did not significantly affect either delay or probability discounting, *p* < 0.05.
Ortner et al. [34]. -Germany -Randomized, between subject design.	Males	91 participants - Experimental group: 46 Placebo group: 45.	Aged 24.3 SD: 2.73	Testogel ^®^ 50 mg Placebo 50 mg	Monetary choice questionnaire [9].	Testosterone does not have a significant effect on delay discounting in male university students, *p* = 0.538.
Petzold et al. [35]. -Germany -Randomized, within-subject design.	Males and females	87 participants -65 males -22 females 44 Low-weight subjects -25 males -19 females	Aged: 20–40 Mean: 35.91 ± 3.8	Madopar 187.5 mg L-dopa Booster 93.75 mg Placebo	Value-based decision-making test battery.	No significant differences between placebo and L-DOPA conditions for delay discounting, risk-seeking for gains and losses, *p* < 0.05.
Pine et al. [36]. -London -Randomized, within-subject design.	Males and females	14 participants -Men: 6 -Women: 8	Aged 18–30 Mean 21	Haloperidol 1.5 mg Madopar (L-Dopa 150 mg) Placebo	Temporal discounting task [37].	L-dopa 150 mg increased the number of sooner options chosen relative to the placebo condition. There was no significant difference between haloperidol and placebo conditions on this disposition, *p* < 0.05.
Reynolds et al. [13]. -United States -Randomized, within-subject design.	Male and female	35 participants -Men 19 -Women 16	Aged 18–45	Diazepam (Valium, 5 mg) Diazepam 10 mg Placebo	Delay discounting task [23].	Diazepam (5 mg & 10 mg) did not affect performance on any of the behavioral task measures of impulsivity, including k values on the discounting task. *p* < 0.05.
Riis-Vestergaard et al. [2]. -Netherlands -Randomized, between-subject design.	Males	79 participants	Aged 18–35	Hydrocortisone (10 mg; rapid and slow cort) Placebo	Intertemporal choice task.	Hydrocortisone administration increased preference for immediate rewards after administration, but not three hours later, *p* < 0.05.
Wagner et al. [38]. - Germany -Randomized, between subject design	Males	54 participants: 27 for each group.	Placebo Group: 24.4. SD: 3.4 - Experimental group: 23.3 SD: 2.5	Haloperidol 2 mg Placebo	Temporal discounting task.	The D2-receptor antagonist haloperidol attenuated temporal discounting and substantially shortened nondecision times.
Weber et al. [39]. - Switzerland - Randomized, between subject design	Male and Female	121 participants - Experimental group: 81 Placebo group: 40	Placebo group: 22.15 Amilsupride group: 21.46 Naltrexone group: 21.65	Amisulpride 400 mg Naltrexone 50 mg Placebo	Monetary Choice Questionnaire [9].	Amisulpride group chose the smaller immediate rewards significantly less often than the placebo group (t [40] = 2.58, *p* < 0.01). The difference between the naltrexone and the placebo groups did not reach significance (t [40] = 1.70. *p* = 0.09).
Wu et al. [41]. -China -Between subjects, placebo-controlled, double-blind intervention.	Males	111 participants	Aged 18–27 Mean: 21.7 ± 1.9	Androgel (150 mg) ^®^. Testosterone Placebo	Intertemporal choice (ITC) task.	Testosterone group (k value: M = 0.10, SD = 0.21) showed increased impulsivity than those in the placebo group (k value: M = 0.022, SD = 0.030), *p* = 0.001.
Zacny & de Wit [42]. -United States -Randomized, within-subject design.	Males and Females	12 participants -6 Men -6 Women	Aged 21–39 Mean: 25.3 ± 3.6	Oxycodone (5 mg, 10 mg & 20 mg). Placebo	Delay and probability discounting task (DPD [9]).	There were no discernible trends of an effect of oxycodone on the DPD.

**Table 2 healthcare-11-01046-t002:** Used Drugs.

Drug	Properties	Uses
Amisulpride	Amisulpride is a dopamine D2 receptor antagonist. At low doses Amisulpride blocks presynaptic dopamine D2 and D3 receptors, increasing dopaminergic levels in the synaptic cleft.	Used in schizophrenia, and to prevent and treat postoperative nausea and vomiting in adults
Pramipexole	Pramipexole is a non-ergot dopamine agonist. It shows specific and strong activity in D2 and D3 receptors.	Used to treat symptoms of Parkinson’s disease and Restless Legs Syndrome
Haloperidol	Haloperidol binds to dopamine D2 receptor. Blocking approximately 60–80% of D2 receptors in the brain. But it also has activity at a number of receptors in the brain.	Used to treat schizophrenia. Also, symptoms of agitation, irritability and delirium
Metoclopramide	Metoclopramide is a dopamine D2 antagonist. It inhibits dopamine D2 and serotonin 5-HT3 receptors in the chemoreceptor trigger zone located in the area postrema of the brain.	Used to treat gastroesophageal reflux disease, and prevention of nausea and vomiting
Levadopa	Levadopa is a dopamine precursor. While dopamine cannot cross the blood brain barrier, Levadopa is able to. After crossing the barrier Levadopa is converted to dopamine	Used in the treatment of Parkinson disease
Bupropion	Bupropion is a norepinephrine/dopamine reuptake inhibitor (NDRI). It inhibits the enzymes involved in reuptaking norepinephrine and dopamine prolonging their action. Specifically, bupropion binds to the norepinephrine transporter (NET) and the dopamine transporter (DAT)	Used in the treatment of major depressive disorder and seasonal affective disorder
Tolcapone	Tolcapone is a selective and reversible inhibitor of catechol-O-methyltransferase (COMT) -COMT is an enzyme responsible for the degradation of catecholamines-.	Used as adjunct therapy in the symptomatic management of idiopathic Parkinson’s disease.
D-amphetamine	D-amphetamine is a noncatecholamine, sympathomimetic amine that acts as Central Nervous System stimulant.	Used in the treatment of attention deficit hyperactivity disorder and narcolepsy.
Yohimbine	Yohimbine is a pre-synaptic alpha 2-adrenergic blocking agent. It increases norepinephrine release	Yohimbine is found in supplements.
Propranolol	Propranolol is a non-selective beta-adrenergic receptor antagonist. Means that it does not have preference for Beta receptors. It inhibits sympathetic stimulation of the heart. It reduces resting heart rate, cardiac output, blood pressure.	Used to treat hypertension, angina, atrial fibrillation, myocardial infarction, migraine, essential tremor, hypertrophic subaortic stenosis, and pheochromocytoma.
Hydrocortisone	Hydrocortisone, or cortisol, is a glucocorticoid secreted by the adrenal cortex. It is essential for life and supports many important cardiovascular, metabolic, immunologic, and homeostatic functions	Used to treat corticosteroid-responsive dermatoses, endocrine disorders, immune conditions, and allergic disorders.
Testosterone	Testosterone is a steroid sex hormone	Used to treat hypogonadism or breast carcinoma in women,
Marinols (Dronabinold)	Marinol (Dronabinol) is a cannabinoid. It’s a synthetic form of THC, which is an ingredient found in marijuana.	Treatment of anorexia and weight loss in people with acquired immunodeficiency syndrome. Also is used in treatment of nausea and vomiting from chemotherapy
Diazepam	Diazepam is a benzodiazepine that binds to receptors in the brain and spinal cord. This binding increases the inhibitory effects of GABA (gamma-aminobutyric acid). GABA is involved in sleep induction, memory, anxiety, epilepsy and neuronal excitability.	Used to treat panic disorders, severe anxiety, and seizures.
Oxycodone	Oxycodone is an opioid. It binds to the mu opioid receptor, and to the kappa and delta opioid receptor.	Used in the treatment of pain.
Naltrexone	Naltrexone is a opiate antagonist. It may block the effects of endogenous opioids.	Used in opioid overdose.

Note. Pharmacodynamics of each drug used, and the use related to medical condition.

**Table 3 healthcare-11-01046-t003:** Drug’s mechanism of action.

Drug	Mechanism	Study
Amisulpride	Dopamine D2 receptor antagonist.	Weber et al. [39]
Haloperidol	Dopamine antagonist.	Wagner et al. [38]
		Pine et al. [36]
Metoclopramide	Dopamine D2 receptor antagonist.	Arrondo et al. [26]
Pramipexole	Dopamine agonist.	Hamidovic et al. [29]
Levodopa	Dopamine agonist.	Pine et al. [36]
		Petzold et al. [35]
D-amphetamine	Indirect agonist of dopamine and norepinephrine.	Acheson et al. [25]
		de Wit et al. [28]
		de Wit et al. [28]
Bupropion	Norepinephrine/dopamine-reuptake inhibitor.	Acheson et al. [25]
Tolcapone	Inhibitor of catechol-O-methyltransferase (Enzyme responsible for the degradation of catecholamines).	Kayser et al. [31]
Hydrocortisone	Glucocorticoid (Cortisol).	Cornelisse et al. [27]
		Riis-Verstergaard et al. [2]
Yohimbine	Alpha 2-adrenergic blocking agent.	Herman et al. [30]
Propranolol	Nonselective β-adrenergic receptor antagonist.	Lempert et al. [32]
Testosterone	Acts on Androgen receptor.	Wu et al. [41]
		Ortner et al. [34]
Oxycodone	Opioid agonists.	Zacny et al. [42]
Naltrexone	Opioid antagonist.	Weber et al. [39]
Marinols	(THC).	McDonald et al. [33]
Diazepam	Benzodiazepine. Increases the inhibitory effects of gamma-aminobutyric acid (GABA).	Reynolds et al. [13]
		Acheson et al. [22]
		Acheson et al. [22]

Note. The table presents the effect of the drug according to the study conducted. The colors represent the intensity of the effect in the temporal discounting: Later reward (blue), Sooner reward (yellow) and No effect (green).

## Data Availability

Not applicable.

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
