# Peer review of "Pharmacological Modulation of Temporal Discounting: A Systematic Review"

_healthcare, 2023, doi:10.3390/healthcare11071046_

Round 1

Reviewer 1 Report

Comments :

Comment-1: Few references cited in this review are older than 10 years. Dear Authors please cite references of articles published with in last 5 years.

Comment-2: Authors mention that The EndNote® removed the duplicates references first automatically; then, the authors did it manually. Which version of Endnote is used here ?

Comment-3:Table 1. Summary of characteristics and results of included studies. This table summarizes most of the studies from USA is there any correlation for citing more studies from USA and fewer studies from rest of the world.

Comment-4: in 2.3. Data Analysis of this review authors capitalize the words unnecessarily and it should be corrected in the revised submission.

Comment-5: Box 1. Used Drugs can me made simple and easy for readers instead of large text n paragraphs.

Comment -6: Box 2 also looks vague with odd colors. Change it in the revised version.

Comment -7 : There are grammatic and punctuation errors all over the manuscript.

for example in 4.2 D2 receptor: Authors wrote  It should be notice that this difference could  be to the differences between the doses used in both studies: 2 mg. (Wagner et al., 2020)  and 1,5 mg.

Reviewer 2 Report

Thank you for giving me this opportunity to review it. The topic is interesting and search strategy is sound. Although, the manuscript writing style must be improved before publication. My detailed comments:

1. Results should be mentioned in more details in the abstract.

2. Key-words should be selected from MeSH.

3. Table-1 should be summarized. For instance, I suppose that 'Statistical
analysis
' is not necessary here. In addition, order of the studies is not clear. Alphabetically or year? Furthermore, citation should be added to the studies in the table.

4. Citation is not needed after this sentence: 'nineteen studies fulfilled the inclusion criteria and went through the quality review.'

5. The manuscript needs exact proof-reading. For instance, a lot of extract dots are seen.

6. Ordering of Box-1 is not clear as well. On the other hand, why those were named 'Box'? They are tables. In addition, citation for any description should be added.

7. Figure-3 ordering should be organized as well.

8. Sub-sections of the Discussion should be rewritten. For instance, D2&3 receptors are sub-types of dopaminergic system and some of drugs are dopamine-modulators. These sub-section system is confused.

9. Limitations and suggestions for future studies should be mentioned substantially.

Round 2

Reviewer 2 Report

Thank you for the revisions. Only one comment still remained. This comment should be addressed in the text not abstract:

Limitations and suggestions for future studies should be mentioned substantially
